# MALDI-TOF MS Profiling and Its Contribution to Mosquito-Borne Diseases: A Systematic Review

**DOI:** 10.3390/insects15090651

**Published:** 2024-08-29

**Authors:** Monique Melo Costa, Vincent Corbel, Refka Ben Hamouda, Lionel Almeras

**Affiliations:** 1Unité de Parasitologie et Entomologie, Département de Microbiologie et Maladies Infectieuses, Institut de Recherche Biomédicale des Armées, 13005 Marseille, France; mcosta.monique@gmail.com (M.M.C.); refkabenhamouda10@gmail.com (R.B.H.); 2Aix Marseille Univ, SSA, AP-HM, RITMES, 13005 Marseille, France; 3IHU Méditerranée Infection, 13005 Marseille, France; 4Institut de Recherche pour le Développement (IRD), MIVEGEC, Univ. Montpellier, CNRS, IRD, 911 Av. Agropolis, 34394 Montpellier, France; vincent.corbel@ird.fr; 5Laboratório de Fisiologia e Controle de Artrópodes Vetores (Laficave), Fundação Oswaldo Cruz (FIOCRUZ), Instituto Oswaldo Cruz (IOC), Avenida Brasil, 4365 Manguinhos, Rio de Janeiro 21040-360, Brazil

**Keywords:** biotyping, life traits, mass spectrometry, mosquitoes, surveillance, vectors

## Abstract

**Simple Summary:**

The deadliest animal in the world is, by far, the mosquito, which spreads numerous viral and parasitic infectious diseases. To control mosquito populations and improve surveillance and personal protective measures, the establishment of rapid, simple, and economic strategies to characterize mosquito fauna and its specific life traits is essential for effective vector management. The present study synthesizes existing evidence on the application of an innovative approach—matrix-assisted laser desorption/ionization–time-of-flight mass spectrometry (MALDI-TOF MS) profiling/biotyping—for the identification of mosquitoes and the analysis of certain life traits, such as vector species, blood-feeding sources, pathogenic agents within the mosquito, and its susceptibility to insecticides. The reliability, low cost, and high-throughput capacity make the MALDI-TOF MS profiling a valuable tool for monitoring and controlling mosquito-borne diseases. These findings, based on this innovative approach, have important implications for public health stakeholders in improving mosquito identification, surveillance, and management of vector-borne diseases.

**Abstract:**

Mosquito-borne diseases are responsible for hundreds of thousands of deaths per year. The identification and control of the vectors that transmit pathogens to humans are crucial for disease prevention and management. Currently, morphological classification and molecular analyses via DNA barcoding are the standard methods used for vector identification. However, these approaches have several limitations. In the last decade, matrix-assisted laser desorption/ionization–time-of-flight mass spectrometry (MALDI-TOF MS) profiling has emerged as an innovative technology in biological sciences and is now considered as a relevant tool for the identification of pathogens and arthropods. Beyond species identification, this tool is also valuable for determining various life traits of arthropod vectors. The purpose of the present systematic review was to highlight the contribution of MALDI-TOF MS to the surveillance and control of mosquito-borne diseases. Published articles from January 2003 to August 2024 were retrieved, focusing on different aspects of mosquito life traits that could be determinants in disease transmission and vector management. The screening of the scientific literature resulted in the selection of 54 published articles that assessed MALDI-TOF MS profiling to study various mosquito biological factors, such species identification, life expectancy, gender, trophic preferences, microbiota, and insecticide resistance. Although a large majority of the selected articles focused on species identification, the present review shows that MALDI-TOF MS profiling is promising for rapidly identifying various mosquito life traits, with high-throughput capacity, reliability, and low cost. The strengths and weaknesses of this proteomic tool for vector control and surveillance are discussed.

## 1. Introduction

Mosquitoes are insects that belong to the Culicidae family. Currently, a total of 3586 mosquito species have been identified worldwide, 88 of which are considered important vectors of human diseases [1]. The primary mosquito vectors belong to the following genera: *Anopheles*, *Aedes*, and *Culex* [1]. Mosquito-borne diseases (MBDs) cause major outbreaks in human populations and account for about 700,000 deaths per year [2]. Malaria is responsible for more than half of these annual mortalities (n = 400,000), followed by dengue (*n* = 40,000) [2]. Mosquitoes are considered as the deadliest animals on Earth [3].

Tropical and subtropical areas are the most affected, but the intense worldwide transportation of people and goods as well as global warming have promoted the spread of invasive vectors in new areas [4,5]. In Europe, incursions of *Aedes* (*Ae.*) *albopictus*, one of the primary vectors of arboviruses, were reported in 26 countries [4,6,7,8] (https://www.ecdc.europa.eu/en/disease-vectors/surveillance-and-disease-data/mosquito-maps (accessed on 12 August 2024). This mosquito species has been incriminated in local transmission of chikungunya in Italy [9,10], as well as chikungunya [11,12], dengue [13,14], and Zika [15] in France. The major dengue vector, *Ae. aegypti* is also expanding its distribution worldwide and is now present in Southern Europe (Madeira Island, Turkey, and Cyprus) [16] as climate change has created more favourable conditions for the vector to survive and thrive [17]. The prevention of human–vector contact (e.g., nets and repellents) and the control of mosquito populations are the main strategies applied to limit MBD outbreaks. Chemical control with insecticides is the most widely used method to manage and prevent the spread of mosquitoes. However, the intense use of the few available compounds over decades has led to the selection of insecticide-resistant mosquito populations [18,19,20,21], reducing the effectiveness of chemical-based vector control.

Vector surveillance is a key component of all vector control programs. Its main goals are to (i) rapidly identify any changes in vector density, diversity, distribution, and insecticide resistance; (ii) assess spatial and temporal risks of pathogen transmission; and (iii) guide timely decisions for vector control [22,23,24]. Rapid and accurate identification of vectors can prevent the establishment of invasive species in new territories by ensuring quick and appropriate vector control interventions [25]. Additionally, maintaining an inventory of local mosquito fauna and monitoring its spatio-temporal evolution are of primary importance for management programs.

Morphological identification remains the conventional method for mosquito species classification [26]. It involves examining the morphological structures of the specimen using dichotomous keys [26]. Morphological identification is a cost-effective method that can be performed in the field. Although it remains one of the most widely used methods for mosquito identification, morphological identification is time-consuming and requires entomological expertise, which has been declining over the last 40 years [27]. Furthermore, specimens can be damaged during sampling, transport, or storage, leading to incomplete species identification due to the loss of essential morphological features [28,29]. To overcome the limitations of morphological identification, molecular techniques based on target gene sequencing have been used as a valuable complementary approach. In general, these molecular methods involve comparing the nucleotide sequences of a molecular marker (i.e., DNA barcode) containing a unique genotypic feature for the analysed species from a mosquito specimen with the known reference sequences available in genomic databases (e.g., GenBank, National Centre for Biotechnology Information (NCBI); Barcode of Life Data Systems (BOLD) database). The accuracy of species-level identification is directly linked to the degree of sequence matching between the query and the database (e.g., proportion of sequence homology/identity and scoring of accuracy) [30,31]. The mitochondrial cytochrome c oxidase subunit I (*COI*) is the commonly used barcode gene for mosquito identification, but additional genes such as the internal transcribed spacer (*ITS2*) are sometimes needed, particularly for sibling or complex species. Despite its important contribution to a reliable mosquito identification and the greater accessibility of this technology over time, these molecular techniques remain relatively expensive, and the specimens belonging to species complex are not always unambiguously identifiable [32,33]. The availability of DNA sequences from interest markers is another crucial factor for successful specimen identification.

In this context, matrix-assisted laser desorption/ionization–time-of-flight mass spectrometry (MALDI-TOF MS) has emerged as an alternative method for vector and pathogen identification. This technology was initially developed in the 1980s by Franz Hillenkamp and Michael Karas, who established a soft desorption ionization of particles using an organic compound called a matrix, giving rise to the name of the technique [34]. The principle of MALDI-TOF MS involves the ionization of sample particles mixed with a matrix using a laser beam. The energy of the laser desorbs the matrix, which transfers ions to sample molecules. These particles are accelerated in a tube under vacuum, and the time required for these particles to travel through the tube to a detector (the time of flight, TOF) determines their mass-to-charge ratio [34]. Finally, the detector transforms the energy measured for each ionized molecule into an electric signal and displays a spectral profile, which primarily represents the most abundant, readily ionized small proteins and peptides with low masses (i.e., ranging generally from 2 to 20 kDa). These spectral profiles serve as a fingerprint of the sample and can be used for biotyping. Comparing these protein profiles with a reference database of spectra created from known samples is essential for classification [34]. Due to its simplicity, speed, and high reliability, MALDI-TOF MS profiling has been routinely used for the identification of microorganisms since the 2000s [29].

The first articles reporting the use of MALDI-TOF MS for arthropod identification were published in 2005 and focused on identifying fruit flies, i.e., *Drosophila melanogaster* [35], and aphid species [36]. Three years later, Dani and collaborators reported the differentiation of *Anopheles* (*An.*) *gambiae s.s.* gender based on the comparison of their antennae MS profiles [37]. It took an additional five years to see the first report of successful mosquito identification using MALDI-TOF MS, which included the three major vector genera: *Aedes* spp., *Anopheles* spp., and *Culex* spp. [38,39]. These studies achieved mosquito identification at the species level, even among specimens from *An. gambiae* complex. Since then, this tool has been successfully used to identify mosquitoes reared in the laboratory or caught in the field, confirming its efficiency in identifying various mosquito species [40,41,42,43]. This success led the scientific community to extend the use of MALDI-TOF MS profiling to assess other mosquito life traits such as the blood meal status [44,45,46,47], population age and gender [48], pathogen infection [49,50], geographical origin [41,42,51], and the effects of mosquito sample storage and preparation [42,52,53,54]. More recently, MALDI-TOF MS has been used to identify mosquito specimens throughout their entire life cycle (i.e., egg, larva, pupa, and adult stages) [38,39,55,56] and to analyse different mosquito body compartments (i.e., head, thorax, and legs) to improve the sensitivity/accuracy of mosquito identification and/or to distinguish cryptic or close-related species [42,57,58].

Overall, MALDI-TOF MS offers several advantages. The most significant is the low cost of reagents, estimated at USD 1–2 per sample [41,59]. Another advantage is that MALDI-TOF MS does not require specialized entomological skills and knowledge. Additional benefits include the specificity per mosquito body part, the low sample volume requirements, and the fast sample processing. The main limitation to the broader application of MALDI-TOF MS profiling in medical entomology remains the high cost of the machine, around USD 200,000, and its maintenance [41]. Despite the high initial investment, once acquired by high-throughput facilities, a fast return on investment is expected. Moreover, the widespread application of this technology in microbiology laboratories for routine identification of microorganisms such as bacteria, fungi, and yeasts has contributed to its acquisition by numerous laboratories, excepted in low–middle-income countries [29].

The aim of the present study is to review the current applications of MALDI-TOF MS profiling to analyse various mosquito life traits related to medical entomology and infectious diseases and to highlight the benefit that this tool may offer for the surveillance and control of mosquito-borne diseases.

## 2. Methodology

The bibliographical search was conducted following the Preferred Reporting Items for Systematic Reviews and Meta-Analyses (PRISMA) guidelines [60,61]. Scientific articles were retrieved from three publication databases: PubMed, Web of Science, and ScienceDirect. The searches were initially performed in February 2023 and updated manually in September 2023. The searches were carried out using specific search terms and their synonyms with the Boolean operator. In PubMed and Web of Science, the search term was limited to “MALDI” AND “Mosquito”. Applying the same terms to ScienceDirect database resulted in retrieving five times more articles (n = 676, with filters applied) than from the PubMed and the Web of Science. An overview of article titles revealed that the majority were outside of the scope of the search terms “MALDI” and “Mosquito”, highlighting that these terms were not well suited for ScienceDirect and that more specific terms should be used. The revised search terms included “MALDI” or “mass spectrometry” combined with terms related to specific mosquito life traits (e.g., “MALDI” AND “mosquito identification”; “mass spectrometry” AND “mosquito identification”, etc.). The complete set of search terms with the respective results of articles retrieved per publication databases is provided in the Appendix A.

Following the PRISMA methodology, the initial screening process for articles to be included in this study was performed as mentioned above. Additional filters were applied, including the publication year, language, and type of article. Duplicate records were then manually excluded, and the remaining articles were screened based on their titles and abstracts to evaluate whether the subject specifically related to the use of MALDI-TOF MS, mosquito and life traits, and/or mosquito control/monitoring/surveillance. The full text of the selected articles was retrieved for further analysis. Other relevant articles that were not retrieved from the databases or those published after the date of the initial search (i.e., between March 2023 and August 2024) were included manually to generate the most up-to-date review.

The initial literature search was not time-limited, but since the technology related to MALDI-TOF MS profiling was developed from the 2000s, only peer-reviewed papers published in English from 2003 to February 2023 were selected from the databases. Review articles, conference proceedings, abstracts without the full text, case reports, case series, non-peer-reviewed documents, and doctoral theses were excluded. Duplicates were also excluded. The assessment of risk of bias was performed by two of the authors of the present review through independent screening of the article titles and abstracts. In case of disagreement, a third senior author was consulted to decide whether to include an article.

## 3. Results and Discussion

A total of 722 articles were initially selected from the repositories, including 134 from PubMed, 131 from Web of Science, and 457 from ScienceDirect. Among these articles, 200 were duplicate records and were excluded. The application of exclusion criteria using filters (i.e., year of publication, language, conference proceedings, and abstracts only) led to the exclusion of an additional 144 articles. Of the 378 remaining articles, 322 were excluded after the first screening based on their titles and abstracts, as they were not relevant to the subject.

The assessment of risk of bias was conducted during the first screening. Two authors classified the articles using the following codes: not included (0), included (1), or maybe (2). In case of doubt or disagreement, a third author (senior researcher) intervened in the discussion. Of 378 articles retained during the first screening step, only 83 (i.e., 22%) articles required a joint analysis of the authors to decide whether they would be included in the present review. The full text of the remaining 56 papers was retrieved, read, and analysed. During this second screening, six articles were excluded because they did not meet the inclusion criteria. Additionally, four articles published after February 2023 that met the inclusion criteria were added manually, resulting in the final inclusion of 54 studies (i.e., 50 from database search and four added manually). The detailed flowchart illustrating the steps of article selection is presented in Figure 1.

Among the mosquito life traits investigated, “mosquito identification” was the most commonly studied. The peak of articles relating to the use of MALDI-TOF for mosquito analysis occurred in 2019 (Figure 2). France has made the largest contribution to the use of MALDI-TOF MS profiling for entomology studies over the last 20 years.

The 54 articles included in the present study were then analysed and sorted according to the mosquito life trait examined: (i) species identification, (ii) vector surveillance (iii) identification of developmental stages, (iv) mosquito life expectancy, (v) tropic preference, (vi) geographic origin, (vii) pathogen infection, (viii) mosquito microbiota, and (ix) mosquito resistance to insecticides. These comprise the following sections, which are completed by a description of promising innovative applications of MALDI-TOF MS for entomological studies.

### 3.1. Assessment of MALDI-TOF MS Profiling for Species Identification

Due to the complexity and time required to correctly identify mosquito species using morphological method and DNA barcoding approaches, MALDI-TOF MS has emerged as a promising tool for systematic studies [26]. As mentioned earlier, the first application of MALDI-TOF MS profiling on mosquitoes occurred in 2008, focusing on the differentiation of *An. gambiae* gender based on MS spectra obtained from their antennae [37]. Since this pioneering work, other studies for mosquito identification have been undertaken, starting in 2013 [38,39]. These two independent studies demonstrated that MALDI-TOF MS could reliably identify adult mosquitoes. Müller and collaborators [38] then evaluated the capacity of MALDI-TOF MS to differentiate 12 distinct *Anopheles* species, primarily from laboratory origin, including members of the *An. gambiae* complex. MALDI-TOF MS successfully differentiated these mosquito species using spectra obtained from the cephalothorax. However, the separation of sibling species was incomplete based on hierarchical cluster analysis of the spectra. The authors failed to identify unique peaks that could serve as single biomarkers to distinguish these closely related species. To overcome the limitations of unsupervised cluster analysis, a supervised statistical model was applied to these morphological similar species. The application of this model on the *An. gambiae* complex (i.e., *An. gambiae s.s.*, *An. arabiensis*, *An. merus,* and *An. quadriannulatus*) demonstrated that 95% of laboratory-reared mosquitoes could be correctly classified. The same model applied to the M and S molecular forms of *An. gambiae s.s.* (the “M form” is now named *An. coluzzii*, while the “S form” retains the name *An. gambiae* Giles) succeeded in correctly classifying 91% of the specimens according to their molecular forms. This work demonstrated that MALDI-TOF MS could be used to discriminate anopheline mosquito species, even at the sibling-species level, with relatively high accuracy [38].

In the same year (i.e., 2013), Yssouf A. and colleagues assessed the performance of MALDI-TOF MS for mosquito identification using specimens from 20 species across six genera (four *Aedes* spp., nine *Anopheles* spp., four *Culex* spp., one *Lutzia* spp., one *Orthopodomyia* spp., and one *Mansonia* spp.) [39]. In contrast to the study conducted by Müller et al. [38], all the mosquito specimens were collected in the field and originated from La Reunion Island and Senegal. The authors selected legs as the body part for MALDI-TOF MS analysis and obtained reproducible and high-quality MS spectra among specimens from the same species. The inclusion of leg MS spectra from each species or sibling species in the reference database allowed them to correctly identify 100% of the specimens tested. The selection of legs for mosquito identification offers several advantages. The majority of the specimen body parts are preserved and could be used for other purposes, such as detecting pathogens in salivary glands or determining parity rate through ovaries dissection. These initial studies confirmed that MALDI-TOF-MS analysis of protein extracts from mosquito body parts is a suitable method for identifying different specimens even to the level of sibling species [38,39]. These early works also highlighted that different body parts could be used for mosquito identification. Since protein repertory varies according to the body part for samples from the same species, standardization of protocol is however required.

Indeed, the lack of guidelines and standardized protocols for selecting mosquito body parts for MS analysis, including sample preparation and homogenization methods, can impair specimen identification. In the literature, legs were the most commonly used mosquito part for identification [39]. However, others studies have demonstrated that different body parts could be successfully used for identifying imagos, such as the head [38], cephalothorax [38,63], thorax, or legs, either individually [39,51,64,65,66,67] or in pairs [54,57,58,68]. In 2016, Nebbak et al. proposed guidelines for body part selection, protocols of sample preparation, preservation mode, storage duration, and homogenization modes for mosquito identification at both larval and adult stages to facilitate MS spectra data exchange among laboratories [53]. For these demonstrations, two species from the *An. gambiae* complex (*An. coluzzii* and *An. gambiae s.s.*) were used. The reproducibility and stability of species-specific MS profiles were endpoints used to define the optimized conditions. The authors concluded that the legs and whole larva were the most suitable body parts for mosquito identification. They also showed that automatic homogenization methods of the samples were better than the manual ones because they allow testing larger sample sizes without variation in homogenization performance, unlike manual grinding. Moreover, in the automatic modes, the addition of glass powder [53] or glass beads [57] as sample disruptors is recommended, and an optimization of the homogenisation conditions is required according to the apparatus used [53]. Other studies comparing manual and automatic homogenization methods also highlighted the superiority of the automatic mode [42]. The automation and standardization of mosquito sample preparation methods for MALDI-TOF MS analyses overcome the time barrier posed by manual sample preparation and the problems of intra-species spectra heterogeneity.

To assess the storage mode and duration of storing, a kinetic submission of mosquito samples stored under the usual laboratory conditions for different duration of time (one week to six months) to MALDI-TOF MS revealed that freezing at −20 °C or in liquid nitrogen (−196 °C) for up to six months appears to be optimal for the identification of both development stages (adult and larval) [53]. Nevertheless, reliable identification was also obtained for adult mosquito specimens stored at room temperature with silica gel for up to 70 days. This preservation method can be an alternative when freezing the samples is not possible, particularly in the field. Conversely, ethanol preservation was not recommended for testing mosquitoes with MALDI-TOF MS [53].

Another study showed the time that samples remain in a mosquito trap can accelerate the protein degradation, which may impact the outcomes [52]. Additionally, during the transport and storage of biological specimens, some parts of the mosquito can be damaged or lost due to handling, particularly the legs, which are fragile. It was repeatedly reported that the quality of the spectra was largely dependent on the number of legs submitted to MALDI-TOF MS, and reducing the number of legs can impair the identification [52,53,69].

To circumvent the limitations of mosquito identification using legs, more recently, other body parts were assessed. Among them, the thorax appeared as a relevant body part. It presents several advantages; it is not fragile, and this relative large body part contains more proteins than legs, which can generate MS spectra of higher intensity [42]. To improve mosquito identification, the submission of two distinct body parts from the same specimen to MALDI-TOF MS was assessed [54]. As protein repertories differ by mosquito body part and since legs and thoraxes have already been reported as relevant compartments for mosquito identification, it was proposed to independently submit these two body parts (i.e., legs and thorax) from a same specimen to MS. The aim was to double-check the identification and assess the concordance of identification between these two body parts. The application of paired sample submission to MS of mosquito field collected in Guadeloupe Island revealed 100% concordant species identification, in agreement with morphological classification [54]. This double-checking system is particularly relevant when close-related species need to be identified or when the quality of the MS spectra from one body compartment is insufficient (e.g., loss of several legs). This strategy was applied for the distinction of eight *Anopheles* spp. [57] and 13 *Culex* spp. [58], including species that could not be distinguished morphologically. Concordant, correct, and relevant identification was obtained for all spectra from both body parts of these last two studies. Interestingly, one specimen classified as *An. peryassui* using morphological criteria was identified as *An. intermedius* based on MALDI-TOF MS analysis of legs and thoraxes [57]. The confirmation of MS results by DNA barcoding underscored the accuracy of the proteomic classification. It is noteworthy that the paired submission to MS of two body parts per specimen improved the identification rate by increasing the associated identification score and the concordance of the results. This paired MS query may be decisive for the distinction of cryptic species.

Although head, thorax, and legs are suitable for mosquito identification with MALDI-TOF MS, the performance of the identification can vary for the same specimen depending on the body part used [41,42]. Nabet et al. argued that the thorax from mosquitoes was the better body part for specimen identification by MS, claiming that better results were obtained with the mosquito head [41]. These authors reported that one limitation of the use of thorax was the risk of spectra alteration in freshly engorged mosquitoes due to the presence of blood trace in this body part, as observed with some field-collected specimens [41]. In another study, the comparison of MS spectra from different body parts of artificially blood-fed mosquitoes confirmed the detection of characteristic MS peaks of blood in the thorax [42]. However, the same MS peaks indicating the presence of trace amounts of blood were also detected in the spectra generated by the legs and head of the same specimens. These peaks in mosquitoes engorged with human blood were located at 15 138 *m*/*z* and 7568 *m*/*z* of MS profiles [42]. Both these MS peaks were already observed in MS spectra profiles from human blood [44]. The blood traces were then attributed essentially to contamination of other body parts (e.g., thorax, head, or legs), which likely occurred during the dissection of the abdomen from engorged mosquitoes. The absence of these blood-associated MS peaks in the spectra of several thorax samples from engorged specimens as well as the much less frequent detection in leg samples supports this hypothesis. It is interesting to note that the blood-associated peaks detected in MS spectra from different body parts tested did not hamper specimen identification. Nevertheless, this blood contamination could decrease the identification score. According to the authors, the order of preference of the body parts to be analysed to obtain an optimal spectral profile for mosquito identification is as follows: thorax, legs, and head [42]. These findings are in agreement with other recent studies published in this field [54,57,58].

Currently, the main limitation for a widespread use of this proteomic tool for the identification of mosquitoes and their life traits is the absence of a central, freely accessible reference database of MS spectra. To overcome this limitation, it is essential that investigators of each study share their reference spectra database [57,58]. The procurement of a large diversity of mosquito species from distinct origins and developmental stages is necessary to enrich reference spectra database. At present, MS profiles for at least nine genera of mosquitoes, encompassing 67 species, are available. Among these nine genera, the highest numbers of MS profiles have been characterized for mosquitoes from the *Anopheles*, *Culex*, and *Aedes* genera. The other six genera consist of a single species (Table 1). The MS databases include different mosquito developmental stages from eggs to adults, which further highlights the potential of MALDI-TOF MS for studies on mosquitoes.

### 3.2. Assessment of MALDI-TOF MS Profiling for Mosquito Surveillance

The successful use of MALDI-TOF MS profiling for the identification of mosquitoes has led to its application in surveillance programs. In this review, four studies that applied this novel technology in medium- to large-scale monitoring programs were identified [6,25,59,71]. MALDI-TOF MS was selected in these studies due to its ease, rapidity, and low cost in identifying mosquito species as compared to the traditional methods based on morphological identification or DNA barcoding. Although the transmission of pathogenic agents by mosquitoes occurs mainly during the adult stage, monitoring and control were generally carried out during the pre-immature (egg) or immature (larvae) stages. Among the four studies that applied MALDI-TOF MS for mosquito surveillance, three relied on egg collection, while one focused on larval stages.

The studies using mosquito eggs were conducted at the Swiss–Italian border [25,59] and in the United Kingdom [6]. In the British study, MALDI-TOF MS contributed, for the first time, to the successful identification of the invasive *Ae. albopictus* [6]. In addition to morphological identification, scanning electron microscopy and direct sequencing of PCR products of *COI* and *ITS2* genes confirmed the identification of the eggs obtained by MALDI-TOF MS. This work demonstrated the robustness of this proteomic tool to monitor the incursions of *Ae. albopictus* in the United Kingdom.

In southern Switzerland, a surveillance program on *Ae. albopictus* was implemented in 2000 and continued for 13 years [25]. This survey was carried out because the neighbouring areas on the Italian side of the border were considered high-risk for the introduction of *Ae. albopictus*. *Ae. albopictus* was identified for the first time in 2003 in Switzerland, in the canton of Ticino, located along the Swiss–Italian border. Initially, the egg, larvae, and adult mosquitoes were submitted to morphological identification. TMALDI-TOF MS was subsequently used to confirm the identification of the eggs given that the classical morphological identification of mosquitoes collected with ovitraps was considered unrealistic for such a long entomology survey. MALDI-TOF MS succeeded in distinguishing the eggs of *Ae. albopictus* from those of *Ae. geniculatus* despite the fact that these two *Aedes* mosquitoes coexist in the same area, and their eggs are not easily distinguishable morphologically [25]. More than 3000 eggs analysed by MALDI-TOF MS were identified as *Ae. albopictus*. The timely surveillance measures implemented along the Swiss–Italian frontier helped to limit the introduction and spread of this vector in this territory, thereby reducing the risk of arbovirus transmission.

As part of the Swiss–Italian border monitoring programme [59], MALDI-TOF MS was also used to identify the introduction of another invasive species, *Ae. koreicus*. Native to Asia, *Ae. koreicus* is able to transmit the Japanese encephalitis virus and dog heartworm (*Dirofilaria* (*D.*) *immitis*). Identified for the first time in 2013 by MALDI-TOF MS, this invasive mosquito species is spreading to Central Europe [59]. A validated MS database curated at Mabritec SA (Riehen, Switzerland) was used as the reference for mosquito identification. Given the low hatching rate, morphological identification of larvae and adults was unfeasible. MALDI-TOF MS was therefore applied as a more practical alternative to distinguish the new invasive *Ae. koreicus* from *Ae. albopictus*. This surveillance programme showed that, in case of a low hatching rate of an invasive mosquito, MALDI-TOF MS can rapidly identify invasive species and could replace the challenging and laborious process of morphological identification of mosquito larvae.

Finally, in the south of France, MALDI-TOF MS technology was applied during the summer of 2015 to monitor the presence of mosquito larvae in an urban area of Marseille [71]. Among 2418 larvae or pupae submitted to MALDI-TOF, 93.4% (n = 2259) were correctly identified, distinguishing five species. The lower rate of relevant identification was obtained for early instar larvae and/or pupae. The lower protein concentration of the early instar larvae and metamorphosis that occurs at the pupal stage are likely to explain the lower matching rate of MS spectra with those of the references. Interestingly, *Culex* (*Cx.*) *impudicus*, a mosquito species not initially included in the database, was nevertheless detected. This result confirmed the high species specificity of MS spectra. Combining the results of species abundance with twelve physicochemical variables of larval habitats shed light on the association between specific environmental factors and the presence of mosquito species. Collectively, these studies confirmed the usefulness of MALDI-TOF MS for large-scale monitoring of mosquitoes at immature stages and may contribute to improving the efficiency of mosquito control programs.

### 3.3. Assessment of MALDI-TOF MS Profiling for Mosquito Identification According to the Developmental Stages

Numerous studies have demonstrated the successful application of MALDI-TOF MS profiling for identification of adult mosquitoes using different body parts [26,29,73,74]. The only body part of the adult stage that is not recommended for mosquito identification by MALDI-TOF MS is the abdomen [26]. The principal reason is the presence of mammalian host blood and/or digested food in the digestive tract, which produces extraneous or artefactual MS spectra. The spectra associated with the presence of food or host blood usually render the acquisition of species-specific protein signatures impossible.

Some authors have also demonstrated that the application of MALDI-TOF MS for mosquito identification can be extended to the immature stages (i.e., egg, larva, and pupa) [56,59,70,71]. Although the lower protein concentration present in the early instars of larvae (L1/L2) may not allow accurate specimen identification by MS [55,71], when late instars of larvae (L3/L4) are used, the rate of correct and relevant identification exceeds 92%, which is largely sufficient for surveillance programs. For the identification of larvae, the use of the whole body is recommended because the dissection of larvae can be laborious and time-consuming, especially for early stages, and because incomplete dissection may lead to spectral heterogeneity [70]. Since gut content could alter intra-species reproducibility of MS spectra, this parameter was assessed for whole larvae. Different diets given to laboratory-reared mosquito larvae had a minor effect on MS profile patterns, confirming the suitability of using the entire specimen at immature stages [71].

As for the eggs in the context of monitoring programs, the ovitraps used for *Aedes* monitoring are frequently laden with eggs, and the identification of the eggs alongside the adult stage may improve the efficacy of vector surveillance. Schaffner et al. [56] tested whether it was possible to identify a pooled mixture of eggs from different *Aedes* species. Eggs from the same species and pools of ten aedine eggs, including two or three distinct *Aedes* species in different ratios, were assessed by MALDI-TOF. All nine aedine species in a collection of eggs from a single species as well as two or three *Aedes* species in mixed pools of ten eggs were correctly identified by MALDI-TOF MS. A minimum of three eggs per species was necessary in the pools for the identification of the species [56].

One of the problems of mosquito identification by MALDI-TOF MS is the requirement for euthanizing the specimens for MS submission regardless of the life stage. The euthanasia of the specimen impedes the performance of other assays that require live mosquitoes, such as the evaluation of insecticide susceptibility. From this perspective, analysis of mosquito exuviae (i.e., the outer skin shed after a moult during the aquatic stages) appeared as an alternative. Nebbak et al. established the proof of concept for the application of MALDI-TOF MS for species identification using exuviae from the fourth-instar and pupal stages of laboratory-reared *Ae. albopictus* and *Ae. aegypti* [72]. The exuviae of each of these two aquatic stages yielded distinct and reproducible MS spectral profile. All exuviae samples were correctly identified at the species level. Although this body part has been successfully used, a few disadvantages have been identified. The cuticles of exuviae are sturdy, which limits protein extraction from them, resulting in MS spectra with low peak diversity. The rapid degradation of exuviae in the field decreases protein abundance, leading to impaired identification of the resulting MS spectra. Moreover, MS analysis of exuviae is not suitable for determining the mosquito biodiversity history of the larval habitat. On the other hand, the use of exuviae offers some advantages, including double confirmation of species identification using both fourth-instar and pupal exuviae from the same specimens and the preservation of live material for other work.

The preceding sections highlighted that MALDI-TOF MS biotyping is suitable for mosquito identification at different life stages (eggs, larvae, pupae, adult, and exuviae) and using different body compartments, and it can thus effectively contribute to vector surveillance. It is emphasized that it is necessary to follow standardized procedures, from the choice of samples to the methods of sample handling and treatment. These are the key points to obtain a reproducible, specific MS protein profile. The addition of species-specific internal biomarker mass sets as calibrators in the samples, as previously suggested [51,56], considerably improves the performance of this proteomic tool.

### 3.4. Assessment of MALDI-TOF MS Profiling to Determine Mosquito Life Expectancy

The transmission of mosquito-borne diseases is directly associated with vector competence and capacity. Vector competence refers to the ability of the vector to become infected and transmit pathogens, while vector capacity encompasses the components of a vector population that determine its potential to transmit pathogens. Key components include the probability of daily survival and extrinsic incubation period (i.e., the period of pathogen multiplication in the salivary glands of the mosquito after an infected blood meal). In this context, mosquito life expectancy is a critical factor influencing the risk of disease transmission. Mosquitoes that survive longer have a higher probability of becoming infected. For example, to transmit malaria parasites, *An. gambiae* females must be older than 10 days to become infective [75]. The more frequently a female feeds on humans, the higher the probability of her becoming infected by malaria parasites.

Traditional methods to measure the mosquito age include the dissection of ovaries to assess the parous rates, analysis of cuticular lipid profiles by gas chromatography–mass spectrometry [76,77], near-infrared spectroscopy, and, more recently, the measurement of transcription levels of aging marker genes [78,79]. These techniques are labour-intensive, time-consuming, and are currently not suited for high-throughput analysis. In this context, MALDI-TOF MS represents an interesting alternative for age-grading studies.

MALDI-TOF MS was used for the first time to measure mosquito age using spectra derived from cuticular lipids [75]. The findings revealed that adult female *An. gambiae* mosquitoes old enough to transmit malaria parasites have different spectral profiles than the younger females. The MS spectra of cuticular lipid from female adults aged one day, seven to ten days, and fourteen days can be clearly distinguished by MALDI-TOF MS procedures. Mated females aged between seven and ten days exhibited a three-fold increase in signal intensity at 570 *m*/*z* and 655–660 *m*/*z* compared to one-day-old virgin females, with the 570 *m*/*z* signal suggested to be associated with mating. In contrast, a decrease in the signal intensity at 535–545 *m*/*z* was noted with increasing age. Higher abundance of lipids at 670–680 *m*/*z* and 700–710 *m*/*z* was found in 14-day-old virgin females. An accurate analysis of MS spectra with multivariate statistical methods revealed that cuticular lipid profiles could effectively detect differences between males and females or between virgin and mated females. MALDI-TOF MS proved to be efficient in determining the age of adult mosquitoes through the intensity and diversity of cuticular peaks. This tool can therefore contribute to risk assessment in vector control programs, particularly for estimating the life expectancy of adult mosquitoes before and after implementation of vector control interventions.

More recently, Piarroux et al. evaluated the capacity of MALDI-TOF MS to classify the spectral patterns of laboratory-reared *An. stephensi* mosquitoes based on entomological drivers of malaria transmission, particularly the age of specimens [80]. To detect protein fingerprints associated with mosquito age (0–10 days, 11–20 days, and 21–28 days), MALDI-TOF MS was coupled with machine learning algorithms (artificial neural networks, ANNs). This sophisticated bioinformatics analysis successfully detected specific *Anopheles* protein profile changes, allowing an association of spectral patterns with mosquito age, with an accurate prediction rate of 73%. The best results were obtained with MS spectra from the thorax. The predicted age groups were not associated with specific spectral peaks but rather with the variations in peak intensity. More recently, the same team improved the age-prediction reliability of *Anopheles* mosquitoes by optimizing deep learning frameworks used for MALDI-TOF MS spectra analyses [81]. The application of the MALDI-TOF MS tool combined with a machine learning approach to malaria vectors collected in the field succeeded to estimate mosquito age with an error of less than 2 days under optimal conditions.

These studies underscore that the estimation of mosquito age by MALDI-TOF MS profiling requires complex analyses, and it should be performed by analysing either cuticular lipid profiles [75] or protein patterns of the thorax [80]. Predicting mosquito age, particularly in human malaria vectors, remains essential for evaluating the risk of disease transmission and deploying adequate vector control interventions.

### 3.5. Assessment of MALDI-TOF MS Profiling to Determine Mosquito Trophic Preferences

The determination of blood meal sources in mosquito vectors is essential to improve our understanding of host–vector interactions and the risk of pathogen transmission. Traditional methods for identifying the mammalian source of blood meals include serological tests such as precipitin and enzyme-linked immunosorbent assay (ELISA) as well as DNA-sequencing of target genes [45,82,83]. The major limitations of serological tests include the unavailability of antibodies against a wide range of potential hosts and the specificity of those antibodies [45]. Although DNA sequencing methods have been successfully applied to detect host blood meal sources, these approaches still have several limitations. These limitations include the high cost of sequencing assays, the poor quality of a blood sample that may have undergone digestion in the engorged mosquito gut, the difficulties in analysing mixed blood meal sources [45,84], or the reliance on the availability of complete DNA sequence in public databases [47].

The first study aiming to assess the relevance of MALDI-TOF MS for blood meal identification was performed on laboratory-reared *An. gambiae* mosquitoes that were artificially engorged on seven distinct blood sources from vertebrates (human, horse, sheep, rabbit, mouse, rat, and dog) [44]. The mosquito abdomen samples containing blood proteins were submitted to MALDI-TOF MS at different time intervals. Specific MS profiles from engorged mosquitoes were observed according to the host blood source, independently of mosquito species. Kinetic analysis revealed that abdominal protein spectra remained stable up to 24 h post feeding. After this time point, the digestion of blood proteins altered the MS profiles, reducing the accuracy of blood source identification, similar to the challenges faced with target gene sequencing [44].

Since this initial study, two further studies have expanded the reference MS spectra database by testing the blood of 18 additional vertebrates, thereby confirming the specificity of abdomen MS profiles from freshly engorged mosquitoes (i.e., ≤24 h) [45,85]. Interestingly, blood samples from three primates (i.e., *Callithrix pygmaea*, pygmy marmoset; *Erythrocebus patas*, hussar monkey; and *Papio hamadryas*, Hamadryas baboon) were tested, and no mismatch with human blood occurred, supporting the high specificity of the spectra. To date, blood samples from a total of 25 distinct hosts have been tested, and their MS spectral profiles have been deposited in the database. It was also demonstrated that mixed blood meals from mosquitoes that fed on two distinct hosts were also correctly identified by MALDI-TOF MS [47]. The high capacity of this proteomic tool to identify the blood source is important for identifying vectors responsible for the transmission of zoonotic pathogens and determining animal reservoirs. In the field, it is generally recommended to crush mosquito abdomens on Whatman filter papers to stop blood digestion. MALDI-TOF MS analysis of dried blood samples on Whatman filter papers successfully identified the origin of blood meal [46]. MALDI-TOF MS was able to identify mosquito species and their respective host for blood-fed specimens collected in five ecological areas of Mali [65]. The success rate of classification reached nearly 93% (n = 651/701) using a Whatman filter.

More recently, the combination of machine learning algorithms with MALDI-TOF MS approach allowed researchers to distinguish past-engorged mosquitoes from unfed mosquitoes by analysing mosquito thorax or legs with a success rate about 80% [80]. In this study, the origin of blood sources was not investigated, as the aim was to distinguish between females that had laid eggs (i.e., parous specimens) from those that had not (i.e., nulliparous) in order to estimate the risk of disease transmission.

These studies reinforce the relevance of MALDI-TOF MS in determining the feeding patterns of freshly blood-fed mosquitoes. However, further experiments are required to expand the database of the mammalian blood sources for mosquitoes. Determining blood meal sources is important for improving our understanding of human–vector interactions and identifying reservoir hosts.

### 3.6. Impact of the Geographic Origin of Mosquitoes on the MALDI-TOF MS Profiling

A reliable and effective entomological monitoring program requires a comprehensive database of reference spectra that covers relevant mosquito species involved in human pathogen transmission [64]. The reliability of MS identification is linked not only to the specificity of the spectra but also to their reproducibility. Several factors could impair the generation of MS spectra, as presented in earlier sections. These include extrinsic factors such as the method of sample preparation and storage and intrinsic factors such as blood contamination, environmental conditions (e.g., climate, nutrition, microbial exposure, and population size), and genetic background of the specimen collected. Some studies have reported that variations in MS spectra can occur in larval and adult specimens belonging to the same species and processed under the same conditions but collected in distinct geographic areas [51,64,70]. These intra-species variations were attributed to changes in protein content, which could be influenced by environmental factors and/or genetic background [70].

The variations of MS spectra among specimens from distinct geographic origins did not hinder species identification [42,51], with very few exceptions [41]. One study reported that the inclusion of three cosmopolitan mosquito species (i.e., *Ae. aegypti*, *Ae. albopictus*, and *Cx. quinquefasciatus*) in the database of mosquitoes from the Pacific region was sufficient to identify mosquito species from Asia and Africa with relevant scores [5]. The low rate of misidentification (2.0%) further indicated that MALDI-TOF MS is a robust method for mosquito identification at a global scale. Nevertheless, the introduction of specimens from the same geographic areas in the database significantly improved the accuracy of identification scores [41,42]. The available evidence suggests that the creation of region-specific reference MS databases could provide higher confidence scores and improve the quality of mosquito identification.

### 3.7. Assessment of MALDI-TOF MS Profiling to Detect Pathogens in Mosquitoes

Detection of pathogens in mosquitoes is important for identifying hotspots of disease transmission risk and estimating the intensity of transmission. Early detection of pathogens in the vector can lead to timely and cost-effective responses, thereby preventing outbreaks. Few studies have investigated the efficiency of MALDI-TOF MS in distinguishing infected from non-infected mosquitoes. A limited number of published studies have focused on the detection of parasitic agents, including three filarioid helminths, namely *D. immitis* (dog heartworm), *Brugia* (*B.*) *malayi* (Malayan lymphatic filarial worm, an etiologic agent of lymphatic filariasis)*,* and *Brugia pahangi* (lymphatic filarial worm, an etiologic agent of lymphatic filariasis) in *Ae. aegypti* [49], as well as malaria parasites (*Plasmodium berghei*) in experimentally infected *An. stephensi* mosquitoes [50,80]. To avoid bias due to environmental factors, these studies were conducted on laboratory-reared and artificially infected mosquitoes. In each of these studies, several mosquito body parts were tested. The performance of MS classification was assessed by comparing the results from MALDI-TOF MS analysis with DNA-barcoding analysis, the latter being the gold standard. The results showed that the highest performance, i.e., high sensitivity (86.6%, 71.4%, and 68.7% for *D. immitis*, *B. malayi*, and *B. pahangi*, respectively) and high specificity (94.1%), was obtained using the cephalothoraxes [49]. Among 37 MS peaks that discriminated between uninfected and infected *Ae. aegypti*, two MS peaks (4073 and 8847 Da) were specific to *Ae. aegypti* infected with microfilariae regardless of the nematode species or mosquito compartment. These two peaks were then considered as biomarkers for *Ae. aegypti* infected with these microfilariae.

In 2017, Laroche et al. tested the efficacy of MALDI-TOF MS in detecting changes in the protein profiles of *An. stephensi* experimentally infected with *P. berghei* [50]. The MS screening of uninfected and infected *An. stephensi* mosquitoes revealed concordant results (98.8%) with those of sequencing methods. The differences between uninfected and infected groups of mosquitoes were attributed to variations in the intensity of MS peaks rather than to the presence or absence of any specific peaks. Recently, a more accurate prediction (78%) for the classification of *An. stephensi* mosquitoes, whether uninfected or infected with *P. berghei*, was obtained for spectral patterns of the thorax using machine learning algorithms [80]. Although the same experimental model was used in both studies, the differences in the performance could be attributed to several factors, such as the experimental conditions of mosquito infections, the delay between infective blood feeding and specimen sacrifice, the mosquito body part used in MS analyses, and the method adopted for data analysis. So far, no studies have reported the use of MALDI-TOF MS profiling to assess arboviruses in mosquito vectors. These preliminary results need to be further confirmed and tested, notably with mosquitoes collected in the field. For determining mosquito infection status using MALDI-TOF MS, standardization is strongly recommended to achieve comparable and reproducible results. Simultaneous identification of mosquito vector species and detection of their associated viral, bacterial, or parasitic pathogens would represent a significant advancement in entomological diagnosis and in assessing the risk of mosquito-borne disease outbreaks.

### 3.8. Assessment of MALDI-TOF MS Profiling to Study the Microbiota of Mosquitoes

The term “microbiota” refers to a collection of populations of microorganisms (viruses, protists, bacteria, and fungi) in a particular place or time. Over the past decade, interest has grown in the role of microbiota, especially the gut microbiota of mosquito vectors, and in host–parasite interactions [86,87]. In the context of the present review, the term “microbiota” mainly refers to the commensal intestinal bacteria in mosquitoes. During the aquatic stage, mosquito larvae feed on organic detritus, including microorganisms, some of which become part of gut microflora in both larval and adult stages [88]. The mosquito gut microbiota has been shown to be indispensable for larval development and adult mosquito survival. However, the various roles that gut microbiota play in mosquito biology—such as female fertility, adult longevity, immunity, formation of peritrophic matrix, nutrition, and protection from insecticides due to insecticide-degrading gut bacteria—remain largely unexplored [89,90]. Studies in this area may reveal the mechanisms involved in insect immunity and provide insight into the factors that influence vector capacity, competence, and insecticide resistance [91]. This knowledge may, in turn, help design more locally adapted strategies to control the spread of various mosquito-borne pathogens, such as arboviruses and malaria parasites [90,92].

In one of the first studies attempting to isolate and reveal the widest possible spectrum of aerobic and anaerobic bacteria present in the midgut of several mosquito species (*An. gambiae* s.l., *Ae. albopictus*, and *Cx. quinquefasciatus*), investigators conducted a large-scale isolation of bacteria through microbial culturomics [93]. The species identification of both mosquitoes and their intestinal bacteria as well as the bacteria found in breeding water for laboratory-reared mosquitoes was confirmed by PCR sequencing and MALDI-TOF MS. Up to 16 bacterial species belonging to 12 genera were identified in the midgut of *An. gambiae*, 11 species from 8 genera were identified in *Ae. albopictus*, and 5 species from 5 genera were identified in *Cx. quinquefasciatus*. The authors confirmed that most of the isolated bacteria belong to the phyla Proteobacteria and Firmicutes [93]. Although some strictly anaerobic bacteria cannot be isolated in cultures, the large-scale isolation technique of culturomics allowed the isolation of 17 additional bacterial species in the mosquito midgut microbiota that had not been previously reported. Most of the bacteria present in the water were found in the midgut of adult mosquitoes, highlighting the direct influence of larval breeding water in the microbiota composition of mosquitoes.

In subsequent studies, other authors reported similar results using MALDI-TOF MS for bacterial identification in the midgut of *Ae. aegytpi*, *Ae. albopictus*, *Cx. quinquefasciatus, An. arabiensis*, and *An. funestus* that were either reared in the laboratory or caught in the field. However, the authors noted some differences in the numbers and species of isolated bacteria due to environmental factors between laboratory-reared and field-collected specimens [94,95]. In the study conducted by Tandina et al. [93], the majority of the isolated bacteria belonged to the phyla Proteobacteria and, to a lesser extent, Firmicutes. Microbial composition and diversity were not affected by sex, storing condition (fresh vs. preservation), preservative mode, or storage period (up to 3 months) [95]. The authors concluded that MALDI-TOF MS is a valid tool for determining the composition of microbiota in mosquitoes using culturomics.

One of the promising strategies under consideration for vector control relies on the massive and regular release of *Ae. aegypti* adult male mosquitoes artificially infected with the bacteria *Wolbachia*. When *Wolbachia*-infected adult male mosquitoes mate with wild females, the eggs do not hatch due to cytoplasmic incompatibility [96]. *Wolbachia* bacteria invade most tissues in mosquito, including the salivary glands, midgut, muscles, and nervous system, while shortening the mosquito’s life expectancy [97]. To evaluate the impact of this strategy, a monitoring of the presence of *Wolbachia* in *Ae. aegypti* mosquitoes after the release in the field is necessary. In this context, MALDI-TOF MS coupled with artificial intelligence was assessed to distinguish *Wolbachia*-infected from uninfected mosquitoes using the head and thorax [98]. This strategy achieved a high accuracy rate of classification comparable to that of quantitative PCR and even superior to the loop-mediated isothermal amplification. The high-throughput assays coupled with the low cost per sample could be particularly relevant to detect the presence of *Wolbachia* in mosquitoes in the framework of large-scale field trials implemented worldwide.

### 3.9. Assessment of MALDI-TOF MS Profiling to Monitor Insecticide Resistance

Insecticide resistance in mosquitoes is considered by the WHO as a serious threat to any vector control program. Currently, methods to detect insecticide resistance rely on biological, molecular, and biochemical tools, each with its strengths and weaknesses [99]. Adequate surveillance of insecticide resistance requires repeated measurements of mosquito susceptibility to insecticides across multiple (sentinel) sites, and as such, the number of samples to be tested can be extremely high [100]. Consequently, more efficient tools are needed to discriminate between susceptible and insecticide resistant mosquitoes, especially when large-scale or nationwide monitoring programs are implemented. Until recently, there were no reports or publications relating the use of MALDI-TOF MS profiling to identify insecticide resistance in insects. Our team conducted the first study to assess the potential of MALDI-TOF MS to distinguish between susceptible and pyrethroid-resistant *Ae. aegypti* by comparing the protein signatures of legs and/or thoraxes of laboratory and field-caught populations with different levels of susceptibility to deltamethrin [101]. For this study, a susceptible reference laboratory *Ae. aegypti* species (strain BORA, French Polynesia) was compared to three inbred *Ae. aegypti* lines from French Guiana, each with a distinct deltamethrin-resistance genotype/phenotype. Interestingly, a peak at 4870 Da was found to be significantly more abundant in the highly pyrethroid-resistant *Ae. aegypti* population compared to the susceptible ones from either the laboratory or French Guiana. However, further analyses failed to find a positive association between kdr-resistant markers (V410L, V1016G/I, and F1534C) and the discriminant peak (i.e., 4870Da). Although these preliminary results are promising, further work is needed to characterise the peak of interest and to validate it as a marker of deltamethrin resistance in *Ae. aegypti* populations [101]. This work opens new research perspectives in the field of insecticide resistance, with the aim to facilitate the monitoring of insecticide resistance by national programs.

### 3.10. Novel Applications of MALDI-TOF MS: Future Perspectives

MALDI-TOF MS profiling has demonstrated its value in analysing various mosquito life traits using specimens from different species and geographical origin. These classifications were generally performed by spectral matching, which could be insufficient for detecting specific criteria. As previously mentioned, the combination of artificial intelligence and MALDI-TOF MS profiling has considerably improved identification precision, as demonstrated with pathogen infections [80,98]. It is likely that machine learning algorithms will be applied and used extensively in the coming years for the classification of MS spectra derived from mosquitoes, particularly to distinguish complex or sibling species, microbiota composition, and other key biological parameters.

In MALDI-TOF MS, the analysis of intact protein profiling (IPP) presents some limitations, as previously observed with the determination of the blood feeding origin. Specifically, the digestion of blood protein impairs MS spectra matching with those of the reference database, reducing host identification success when processing time exceeds 24 h post blood feeding [44]. To extend the time window for host identification, peptide mass mapping (PMM) or fingerprinting (PMF), which involves analysing host-specific haemoglobin peptides using MALDI-TOF MS, was assessed using blood-fed female of *Phlebotomus* spp. (vectors of *Leishmania* spp.) and laboratory-reared *Culex* mosquitoes [102]. The principle involves digesting the blood from arthropod abdomens by exogenous trypsin, resulting in peptide fragments that serve as unique host signature in MALDI-TOF. Since the half-life of haemoglobin peptides is longer than the respective whole proteins, PMM is less affected by blood degradation, and these tryptic maps allow conclusive host assignment up to 48 h after the blood meal intake. Interestingly, the application of the PMF to distinguish closely related *Culicoides* species strongly improved their classification [103]. The PMM approach appears to be a reliable alternative for studying the origin of blood meals in mosquitoes and could be extended to other characteristics, particularly for identifying of closely related species or for mosquito exuviae studies [72].

MALDI-TOF MS profiling in addition to protein investigations has emerged as a potential alternative tool for the separation and detection of nucleic acids. It has been employed to differentiate genotypes based on the mass of variant DNA sequences [104]. Among the studies related to mosquito life traits, three have combined the detection and characterisation of nucleotide changes by MALDI-TOF MS. The first addressed the detection and typing of dengue virus [105]. The principle involved comparing RNA fragment profiles resulting from digestion by Rnase T1 endoribonuclease of specific dengue virus PCR products, with an in silico database of digestion patterns from dengue strains. This study provided proof-of-concept for classifying hundreds of dengue viruses down to the serotype and strain level, and it revealed the high potential of this strategy for accurate dengue virus biotyping, which could be extended to other arboviruses.

The two other works investigated the efficiency of MALDI-TOF MS in genotyping single-nucleotide polymorphism (SNP) sites in mosquito genes involved in insecticide resistance [106,107]. The principle involves amplifying a gene containing the SNP target and extending the probe adjacent to the SNP site using ddNTPs. The base type at the target site is then determined by the mass of the extended probe using MALDI-TOF MS. The multiplexing of amplified target sites allows the simultaneous detection of multiple mutation sites. Five polymorphisms in the *acetylcholinesterase-1* gene, which is related to insecticide resistance in *Anopheles* spp. and *Culex* spp., were analysed by Mao et al. [106], while three mutations in the gene coding for the voltage-gate sodium channels—the main target of pyrethroid insecticides—resulting in 17 genotypes in *Ae. albopictus* mosquitoes, were investigated by Mu et al. [107]. In both studies, the comparison of the multiplex PCR–mass spectrometry with conventional molecular genomic methods showed consistent genotyping results, confirming the accuracy of this approach. This technology enabled the rapid screening of several mutations in a single reaction per sample, reducing the cost of the analysis to one-fifth [106]. Further application and development of this competitive strategy for rapid and reliable determination of other mosquito life traits based on SNPs may be promising for surveillance programs.

MALDI–mass spectrometry imaging (MALDI-MSI), which uses high-resolution images based on protein profiles, represents another promising application of MALDI-TOF MS in medical entomology [108]. MALDI-MSI is a powerful tool that provides information on chemical composition and the spatial distribution of molecules from a sample sliced and loaded onto a glass slide. The images are created based on the mass-to-charge ratio of ions of interest measured by MALDI-TOF MS. Additionally, MALDI-MSI has the advantages of being highly sensitive, and unlike other classical imaging methods, it can analyse from hundreds to thousands of molecules simultaneously in a single run, without labelling or altering the scanned tissue. This technology opens the door to spatial analysis of a range of analytes, including peptides, proteins, protein modifications, drugs, and their metabolites or lipids. It offers the potential to map specific molecules in arthropod specimens like endosymbiont or pathogen proteins, the distribution of insecticides, and protein repertory adaptation to environmental changes [109]. So far, only one publication has reported the application of MALDI-MSI to investigate the phospholipid composition, distribution, and localization in whole-body sections of the malaria vector *An. stephensi* [110]. Such studies may represent the first step towards further understanding of parasite–host interaction, in particular the lipid biochemistry underlying malaria infection, which in turn may reveal potential drug targets.

## 4. Conclusions

The contribution of MALDI-TOF MS to biological sciences and medicine has been demonstrated for more than 20 years. Here, we demonstrated that this technique has immense potential for application in medical entomology, particularly for entomology surveillance and the identification of various mosquito life traits. MALDI-TOF MS is a versatile and accurate tool that can be used to determine mosquito species, trophic preference, age, and pathogen infections in mosquitoes. The establishment of species-specific MS protein signatures, regardless of the mosquito’s development stages, from eggs to imago including exuviae allows for the identification of mosquito species throughout their entire life cycle. For adult mosquitoes, combining different body parts enhances identification accuracy, which could be decisive for distinguishing sibling species among complexes. This proteomic tool can be highly beneficial for surveillance programs, as it has shown to provide early and rapid identification of invasive vector species. Although MS profiling offers numerous advantages—such as cost effectiveness, speed, and the ability to conduct high-throughput assays suitable for large-scale mosquito monitoring programs—its use in entomology remains relatively undisclosed. The lack of an international reference MS spectral database likely explains the limited interest for this technique despite significant progress in this field. However, the continuous emergence and reemergence of mosquito-borne diseases worldwide highlight the need to develop more sensitive technologies to improve the evaluation, surveillance, and prediction of epidemics. Integrating MALDI-TOF MS profiling into entomological surveillance could contribute to reducing the burden of mosquito-borne diseases and guide decision making for vector control.

## Figures and Tables

**Figure 1 insects-15-00651-f001:**
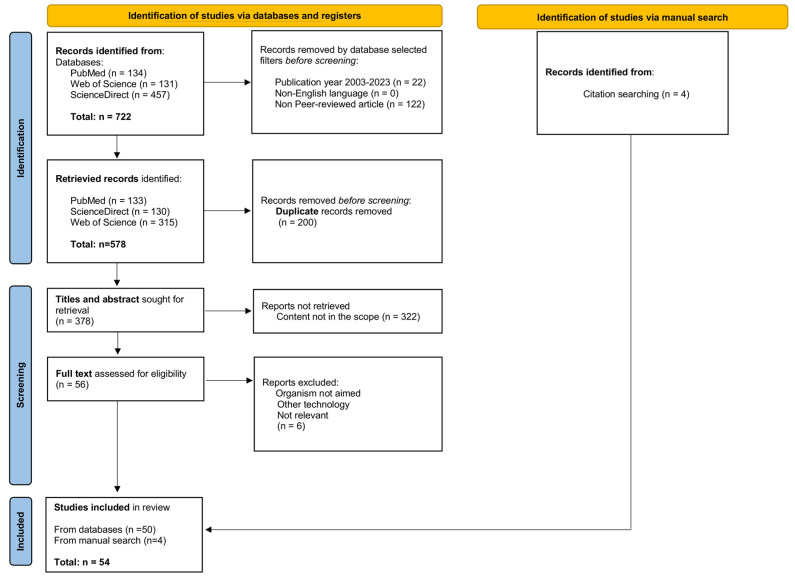
Flowchart showing selection process for a systematic review of the literature. Methodology and detailed results of the search, inclusion, and exclusion of studies followed the PRISMA model [62].

**Figure 2 insects-15-00651-f002:**
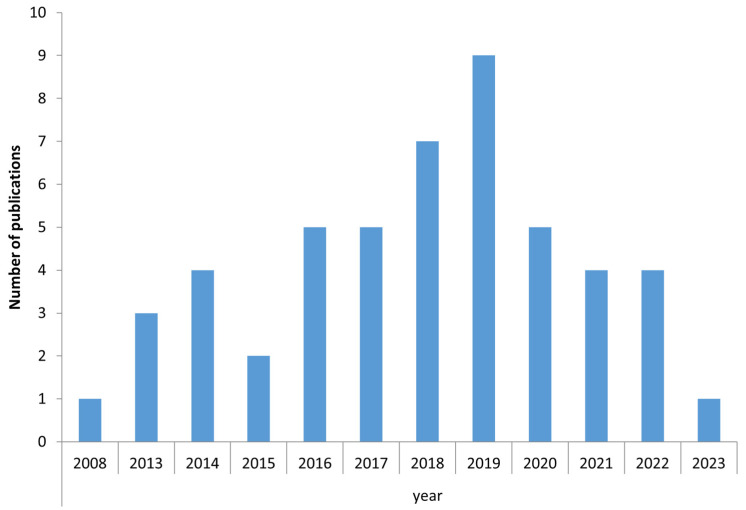
Published articles reporting MALDI-TOF MS and mosquito life traits. Number of publications per year, from January 2003 to September 2023, retrieved from the scientific databases on MALDI-TOF MS profile and mosquito life traits.

**Table 1 insects-15-00651-t001:** Availability of spectral profiles for mosquito species in reference MS database.

Genus (n = 9)	Species (n = 67)	Pre-Immature and Immature Stages	Imago (Compartment)	Reference *
*Anopheles*	*An. aquasalis*		Adult (legs and thorax)	[57]
	*An. braziliensis*		Adult (legs and thorax)	[57]
	*An. darlingi*		Adult (legs and thorax)	[57]
	*An. nuneztovari* (s.l.)		Adult (legs and thorax)	[57]
	*An. triannulatus* (s.l.)		Adult (legs and thorax)	[57]
	*An. oswaldoi* (s.l.)		Adult (legs and thorax)	[57]
	*An. intermedius*		Adult (legs and thorax)	[57]
	*An. peryassui*		Adult (legs and thorax)	[57]
	*An. gambiae*	Larvae, pupae	Adult (head, legs, and thorax)	[39,41,42,53,70]
	*An. coluzzii*	Larvae, pupae	Adult (legs and thorax)	[42,53,70]
	*An. funestus*		Adult (head, legs, and thorax)	[39]
	*An. ziemanni*		Adult (legs)	[39]
	*An. arabiensis*		Adult (head, legs, and thorax)	[39,41]
	*An. wellcomei*		Adult (legs)	[39]
	*An. rufipes*		Adult (legs)	[39]
	*An. pharoensis*		Adult (legs)	[39]
	*An. maculipennis*	Larvae		[71]
	*An. claviger*		Adult (legs)	[64]
	*An. hyrcanus*		Adult (legs)	[64]
	*An. maculipennis*		Adult (legs)	[64]
	*An. bancroftii*		Adult (legs)	[52]
*Culex*	*Cx. dunni*		Adult (legs and thorax)	[58]
	*Cx. nigripalpus*		Adult (legs and thorax)	[58]
	*Cx. quinquefasciatus*		Adult (legs and thorax)	[39,52,54]
	*Cx. usquatus*		Adult (legs and thorax)	[58]
	*Cx. adamesi*		Adult (legs and thorax)	[58]
	*Cx. dunni*		Adult (legs and thorax)	[58]
	*Cx. eastor*		Adult (legs and thorax)	[58]
	*Cx. idottus*		Adult (legs and thorax)	[58]
	*Cx. pedroi*		Adult (legs and thorax)	[58]
	*Cx. phlogistus*		Adult (legs and thorax)	[58]
	*Cx. portesi*		Adult (legs and thorax)	[58]
	*Cx. rabanicolus*		Adult (legs and thorax)	[58]
	*Cx. spissipes*		Adult (legs and thorax)	[58]
	*Cx. nigripalpus*		Adult (legs and thorax)	[54]
	*Cx. pipiens*	Larvae, pupae	Adult (legs)	[39,64,70,71]
	*Cx. modestus*		Adult (legs)	[64]
	*Cx. hortensis*	Larvae		[71]
	*Cx. atratus* (s.l.)		Adult (legs and thorax)	[54]
	*Cx. iyengari*		Adult (legs)	[52]
	*Cx. sitiens*		Adult (legs)	[52]
	*Cx. annulirostris*		Adult (legs)	[52]
	*Cx. molestus*	Larvae and pupae		[70]
*Aedes*	*Ae. aegypti*	Eggs, larvae, pupae, and exuviae of larvae and pupae	Adult (legs and thorax)	[39,42,52,54,56,70,72]
	*Ae. albopictus*	Eggs, larvae, pupae, and exuviae of larvae and pupae	Adult (legs and thorax)	[39,42,53,54,56,70,71,72]
	*Ae. atropalpus*	Eggs		[56]
	*Ae. cretinus*	Eggs		[56]
	*Ae. geniculatus*	Eggs		[56]
	*Ae. japonicus*	Eggs		[56]
	*Ae. koreicus*	Eggs		[56]
	*Ae. phoeniciae*	Eggs		[56]
	*Ae. triseriatus*	Eggs		[56]
	*Ae. taeniorhynchus*		Adult (legs and thorax)	[54]
	*Ae. cinereus*		Adult (legs)	[64]
	*Ae. vexans*		Adult (legs)	[52,64]
	*Ae. caspius*	Larvae	Adult (legs)	[62,68]
	*Ae. rusticus*		Adult (legs)	[64]
	*Ae. excrucians*		Adult (legs)	[64]
	*Ae. scutellaris*		Adult (legs)	[52]
	*Ae. notoscriptus*		Adult (legs)	[52]
	*Ae. vigilax*		Adult (legs)	[52]
*Mansonia*	*M. uniformis*		Adult (legs)	[39]
*Culiseta*	*Cs. longiareolata*	Larvae		[71]
*Deinocerites*	*D. magnus*		Adult (legs and thorax)	[54]
*Psorophora*	*P. cingulata*		Adult (legs and thorax)	[54]
*Coquillettidia*	*Cq. richiardii*		Adult (legs)	[64]

* All the mosquito species identification was carried out by MALDI-TOF MS with a Microflex LT MALDI-TOF Mass Spectrometer (Bruker Daltonics), with the exception of the work realized by Schaffner et al. [56], which used a MALDI-TOF Mass Spectrometry Axima™ Confidence machine (Shimadzu-Biotech, Manchester, UK)).

## Data Availability

The complete set of search terms after selection of filters with the respective results of articles retrieved per publication databases is provided in the Appendix A.

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
