# Peer review of "MALDI-TOF MS Profiling and Its Contribution to Mosquito-Borne Diseases: A Systematic Review"

_insects, 2024, doi:10.3390/insects15090651_

Round 1
Reviewer 1 Report
Comments and Suggestions for Authors
The manuscript describes a systematic review that seems to have been performed well and the manuscript presents the different aspects of how MALDI-TOF MS has been used for mosquito research. In general, this is an informative review but the English is quite poor and needs to be addressed. There is some confusion when describing other species identification methods. From Wikipedia: “Taxonomy is a practice and science concerned with classification or categorization. Typically, there are two parts to it: the development of an underlying scheme of classes (a taxonomy) and the allocation of things to the classes” this should not be confused with morphological identification to species. Molecular identification is not only DNA-barcoding but should include any use of molecular markers, including protein markers as used in MALDI-TOF MS identification. It would be clearer if the manuscript used a more precise language when describing these methods.
I have listed a few of the specific language issues below but there are more sentences that need to be corrected.
” The deadliest animal in the world is by far the mosquitoes” singular/plural
“The present study synthesizes existing evidence on application of an innovative approach, the matrix-assisted laser desorption/ionization time-of-flight mass spectrometry (MALDI-TOF MS) profiling/biotyping, for the identification of mosquitoes and some life traits like vector species, blood feeding source, some pathogenic agents inside the mosquito or it susceptibility to insecticides.” complicated sentence, check grammar.
“The reliability, low cost and high throughput capacity, make of MALDI-TOF MS profiling, a relevant tool for the monitoring and control of mosquito-borne diseases. “ check grammar.
“Currently, morphological and molecular approaches are the standard methods used for vector identification, however, they present several limitations. “ split to two sentences.
“Published articles from January 2003 to August 2024 were retrieved, considering different aspects of mosquito life traits which could be determinant in the transmission of diseases and vector management” Check grammar.
“spatial-temporal evolution” should be, spatio-temporal evolution
“primordial importance” should be, primary importance
“molecular method” is not only DNA-barcoding. Also MALDI-TOF is a molecular method since it compares molecules. If you wish to specifically discuss DNA barcoding or DNA sequencing you should use these terms. In addition. Species specific PCR is another molecular method that has its use when looking for specific mosquito species, such as invasive mosquitoes.
Line 104 ”primordial” should be “primary”
Line 233 “taxonomy method” should be morphological method. Taxonomy is the classification in itself and is not only relating to the morphological methods.
Line 271 “protein repertory” should be changed to ”protein composition”
Line 310 “notably the legs which are breakables”
Line 352 “The blood traces were then attributed essentially to contamination of safe tissues which should occurred during the specimen dissection of engorged mosquitoes.” It is unclear what safe tissues refer to.
Line 594 “molecular test” should presumably be “DNA-sequencing”
Line 607 “vertebrates” all listed species are mammals.
Line 664 “proceed” should be “processed”.
Line 699 “molecular analysis” specify whether this refers to DNA-barcoding or specific PCR or any other technique.
“Assessment of MALDI-TOF MS profiling to monitor insecticide resistance” The possibility to evaluate insecticide resistance is of course of great interest. However, the paragraph lacks a broader description of how insecticide resistance is developed through many different pathways, with point mutations in several different genes. To identify all these different types of insecticide resistance would require a large catalogue of resistant strains to compare spectra from. It would be more informative if the work reviewed is put in this larger picture.
Line 940 “Despite the numerous advantages of MS profiling, including its high cost-effectiveness, rapidity, and high throughput assays compatible with large-scale mosquito monitoring programmes, its use in entomology remains confidential.” This needs to be rephrased.
Comments on the Quality of English LanguageThe language of the manuscript needs to be addressed before publication.
Reviewer 2 Report
Comments and Suggestions for Authors
This manuscript is a systematic review of the application of matrix-assisted laser desorption/ionization time-of-flight mass spectrometry (MALDI-TOF MS) in the determination of several biological parameters of the insects vector of diseases and the identification of pathogens of importance in public health. The built-in of this technique in vector surveillance and control could be useful in the future. This review is very well supported and concretely summarizes the advances in the built-in of this tool and the possible new applications in the field of research. The only drawback is that this technique is not yet very accessible for low- or middle-income countries which are where vector-borne diseases are most prevalent.
Reviewer 3 Report
Comments and Suggestions for Authors
Dear Authors,
The manuscript you wrote represents a valuable source of data and reference guiding to the all major possibilities of MALDI-TOF considering the studies focused on mosquitoes. The material is well-written and it has a good structure. The text is clear and simple and the language is fine. There is nothing negative to criticize in this work. The manuscript represents the significant contribution to the science.
I have several comments and suggestions that need to be addressed before the manuscript is published:
L60 Here should ECDC be added because they have updated maps.
L63 For Ae. aegypti in Cyprus the citation should be added: Vasquez, M.I., Notarides, G., Meletiou, S., Patsoula, E., Kavran, M., Michaelakis, A., Bellini, R., Toumazi, T., Bouyer, J. and Petrić, D., 2023. Two invasions at once: update on the introduction of the invasive species Aedes aegypti and Aedes albopictus in Cyprus–a call for action in Europe. Parasite, 30.
L75 Please correct vector
L82 Taxonomic identification? Or Morphologic identification?
L126 Please delete sensu stricto and the brackets because s.s. is well-known abbreviation. Also, species An. gambiae is mentioned for the first time in text and should not be abbreviated.
L165 Small letters of - AND. You had that few times in that paragraph.
Figure 1 is blurry and the text is not readable. Please replace it.
Figure 2. Blurry. Please improve the quality
The title of Figure 2. has repetitions. Please revise it.
L131 Please abbreviate the name of species when they are mentioned second, third etc. time in text. Same in L236. Please act accordingly in the whole MS for all species.
L240 Missing space- [37]then.
L280 Please delete comma- et al.,
L342 Please correct- was the better body part… to - were the better body parts
Table 1. Please use small letters for the stages and body parts. Novel classification placed all Ochlerotatus in Aedes genus. So, caspius should be Aedes caspius or you should also revise other Aedes species. In the following text you classified eggs as pre-immature stage, but in table is immature. Please change in text or in table. Please also correct imature to immature in the table 1.
L390 Please correct colection to collection.
L392 Missing space after bracket.
L402 No abbreviations at the beginning of the sentence.
L408 Did you mean survey?
L410-412 Eggs of Aedes geniculatus and Aedes albopictus are easily distinguishable for medical entomologists. Please correct this.
L471 You have one redundant bracket. Please delete.
L493 Please delete- (i.e., the outer skin that is shed off after a moult during the aquatic stages)… because this is well-known term in entomology.
L690 Please shorten the D. immitis because you mentioned it before in the text.
L727 Please correct stronglhy
L799 Please correct - straegy
Reference 2. It has 2023 2023. If you would like to keep both, please add comma. Please add link or more details.
Comments on the Quality of English LanguageThe language is fine.
Round 2
Reviewer 1 Report
Comments and Suggestions for Authors
The manuscript is a good introduction to the work that has been done using MALDI-TOF for mosquito surveillance and it deserves to be published.
Comments on the Quality of English LanguageWhile the langage is improved it is still not at a standard that is expected in a published review article. I cannot list all examples but encourage the authors to use as much help as possible to improve the readability of the manuscript. I list a number of examples that I found but there are probably more sentences that should be rephrased. Please take the oppurtunity to go through the manuscript for language errors.
Line 168: ”It consists to research morphological structures of the specimen based on the use of the dichotomous keys [26]”
I suggest that you write: ”Morphological identification remains the conventional method for mosquito species classification [26]. It is based on classification of morphological structures of the specimen following dichotomous keys [26].”
Line 172: . ”In addition, specimen damages could occur during the sampling, transport or storing of mosquitoes. These subsequent alterations may conduct to incomplete species determination due to the loss of essential morphological criteria [28,29].”
I suggest that you write: ”In addition, a specimen may be damaged during the sampling, transport or storing of mosquitoes. Which can lead to incomplete species determination due to the loss of essential morphological criteria [28,29].”
Line 177: ” In general, these molecular method consists in the comparison of the nucleotide sequences of a molecular marker (i.e., 178 DNA barcode), containing a unique genotypic feature for the analysed species, from a mosquito specimen with the known reference sequences, available in genomic databases (e.g., GenBank, National Centre for Biotechnology Information [NCBI]; Barcode of Life Data Systems [BOLD] database).”
Choose to use singular or plural ”these” does not go with ”method”
